# Screening *Vitis* Genotypes for Responses to *Botrytis cinerea* and Evaluation of Antioxidant Enzymes, Reactive Oxygen Species and Jasmonic Acid in Resistant and Susceptible Hosts

**DOI:** 10.3390/molecules24010005

**Published:** 2018-12-20

**Authors:** Mati Ur Rahman, Muhammad Hanif, Ran Wan, Xiaoqing Hou, Bilal Ahmad, Xiping Wang

**Affiliations:** 1State Key Laboratory of Crop Stress Biology in Arid Areas, College of Horticulture, Northwest A&F University, Yangling 712100, China; mati@nwafu.edu.cn (M.U.R.); mhanif@nwafu.edu.cn (M.H.); wanran2012@nwsuaf.edu.cn (R.W.); houxiaoqing09@nwafu.edu.cn (X.H.); bajwa1999@nwafu.edu.cn (B.A.); 2Key Laboratory of Horticultural Plant Biology and Germplasm Innovation in Northwest China, Ministry of Agriculture, Northwest A&F University, Yangling 712100, China; 3College of Horticulture, Henan Agriculture University, Zhengzhou 450002, China

**Keywords:** biotic stress, *Botrytis cinerea*, grape, resistant genotypes, reactive oxygen species, jasmonic acid, antioxidant enzymes

## Abstract

*Botrytis cinerea* is a necrotrophic fungal phytopathogen with devastating effects on many *Vitis* genotypes. Here, a screening of 81 *Vitis* genotypes for leaf resistance to *B*. *cinerea* revealed two highly resistant (HR), twelve resistant (R), twenty-five susceptible (S) and forty-two highly susceptible (HS) genotypes. We focused on the HR genotype, ‘Zi Qiu’ (*Vitis davidii*), and the HS genotype ‘Riesling’ (*V*. *vinifera*), to elucidate mechanisms of host resistance and susceptibility against *B*. *cinerea*, using detached leaf assays. These involved a comparison of fungal growth, reactive oxygen species (ROS) responses, jasmonic acid (JA) levels, and changes in the anti-oxidative system between the two genotypes after inoculation with *B*. *cinerea*. Our results indicated that the high-level resistance of ‘Zi Qiu’ can be attributed to insignificant fungal development, low ROS production, timely elevation of anti-oxidative functions, and high JA levels. Moreover, severe fungal infection of ‘Riesling’ and sustained ROS production coincided with relatively unchanged anti-oxidative activity, as well as low JA levels. This study provides insights into *B*. *cinerea* infection in grape, which can be valuable for breeders by providing information for selecting suitable germplasm with enhanced disease resistance.

## 1. Introduction

*Botrytis cinerea* is a necrotrophic fungal phytopathogen that causes devastating gray mold disease in more than 200 dicotyledonous plant hosts, as well as some monocotyledonous species. This polyphagous pathogen is the second most prevalent phytopathogen responsible for pre-and postharvest decay and fruit quality deterioration in greenhouses, open fields, and during storage, including cold storage (0–10 °C) [1]. Grey mold is a major challenge to grape cultivation worldwide where periods of relative humidity (>90%) and cold temperatures (14–28 °C) persist for a long time and coincide with bloom and ripening [2].

The traditional control of *B*. *cinerea* includes strong fungicide treatments during the seasonal crop cycle, and current strategies for control of *Botrytis* bunch rot rely on a combination of canopy management and fungicide usage. Physical methods, such as the use of fruit zone leaf removal, shoot positioning, and timely training and pruning can reduce disease severity, but these techniques are expensive, laborious, and less effective than fungicides [3,4]. In addition to increasing the cost of grape production, the use of chemical fungicides has an adverse impact on the environment, and potentially on consumer health [5]. There are also reports of fungal strains developing resistance to some commonly used fungicides and this has resulted in restrictions being adopted in the use of chemicals for crop protection [6,7,8]. Thus, the development of resistant, high-quality genotypes would reduce the dependence of the viticulture industry on pesticide input and have significant environmental and economic benefits. The most commonly cultivated species, *V. vinifera*, lacks resistance to many diseases, and the degree of susceptibility varies between cultivars and depends on environmental conditions [9,10].

Previously, Wan et al., evaluated the responses of different wild Chinese *Vitis* genotypes to *B. cinerea* infection. However, the resistance level is highly variable among the clones of the same species, for example, Lueyang-4, Ningqiang-6 and Tangwei (*Vitis davidii*) [11]. In this study we investigated the responses of different *Vitis* genotypes i.e., eleven (*V. vinifera × V. labrusca*), 65 (*V. vinifera),* three (*V. vinifera* L *× V. amurensis* genotypes), one (*V. amurensis),* and one clone of (*Vitis davidii*) to *B. cinerea* infection. The phenotypic and histochemical disease signs and symptoms at different levels of *Vitis*—*B. cinerea* interaction were evaluated in a total of 81 *Vitis* genotypes. In addition, we measured the levels of reactive oxygen species (ROS), which play important roles in plant physiology, including development, cellular signaling, and biotic and abiotic stress tolerance. ROS production needs to be tightly regulated to balance the physiological effects [12]. Substantial evidence indicates that *B*. *cinerea* challenge can overthrow the effects of ROS stress on plants [13,14].

In addition to ROS, the plant hormone jasmonic acid (JA) is known to be involved in biotic stress amelioration [15]. For example, JA plays a key regulatory role in defense responses to necrotrophs [16] and in participating in responses to injury and biotic stresses, such as occur during insect and pathogen attacks [17]. In this study, we investigated the levels of ROS, the antioxidant enzymes, and JA in the leaves of highly resistant (HR) and highly susceptible (HS) *Vitis* genotypes. These results were correlated with disease resistance at different time points post-inoculation with *B. cinerea*, generating information that may assist in future breeding programs.

## 2. Results

### 2.1. Grape Genotypes Exhibit Different Levels of Resistance to B. cinerea

Detached leaf assays were used to evaluate *B*. *cinerea* infection in vitro [18] and the spread of leaf lesions caused by *B*. *cinerea* were measured at 96 hpi (hours post inoculation) (Figure 1). Some genotypes displayed significant differences in *B*. *cinerea* resistance (Table 1), and a completely randomized least significant difference (LSD) analysis showed that the similarity was found among the replicates, and that the average disease severity was significantly different (*p* ≤ 0.05) between the various genotypes (Table 1). No significant difference (*p* > 0.05) was found between the 2016 and 2017 data (Appendix A).

In total, 81 *Vitis* genotypes were evaluated to investigate the resistance level against *B*. *cinerea*. All genotypes were classified according to their disease severity index (SI) at 96 h post-inoculation (hpi). Among the 81 genotypes, 42 (Table 1) were HS according to a disease SI of 5.51–7.0. Mycelium and sporulation was observed on these genotypes. A total of 25 genotypes (Table 1) were S, with mycelium production at 96 hpi, with no/less sporulation (SI of 3.51–5.50). A total of 12 genotypes (Table 1) were resistant, with much less mycelium production and no sporulation was observed in genotypes with SI values of 1.51–3.50. The ‘Zi Qiu‘ and ‘Ju Mei Gui’ genotypes (Table 1) were HR with no mycelium or sporulation and with SI values of 0–1.50.

Two representative genotypes each from the HS, S, R and HR categories were chosen for macroscopic and microscopic evaluation of fungal growth at 96 hpi (Figure 2). The leaves of the HS ‘Riesling’ (Figure 2D,L) and ‘Pinot noir’ (Figure 2H,P) genotypes were completely covered in mold and were roofed by mycelium, as well as showing signs of sporulation. The S genotypes ‘Flame’ (Figure 2C,K) and ‘Canner’ (Figure 2G,O) had numerous mycelia but showed no signs of sporulation. The R genotypes ‘Kang San’ (Figure 2B,J) and ‘Rizamat’ (Figure 2F,N) formed significantly fewer necrotic lesions than the HS and S genotypes. Moreover, conidia with penetrating pegs were observed on the leaves of ‘Kang San’ and ‘Rizamat’; however, the resulting hyphae did not extend, indicating restricted *B*. *cinerea* proliferation. The HR genotypes ’Zi Qiu’ (Figure 2A,I) and ‘Ju Mei Gui’ (Figure 2E,M) had 2% and 8% lesion areas, respectively (Appendix A).

### 2.2. B. cinerea Growth on the HR Genotype ’Zi Qiu’ and the HS Genotype ‘Riesling’

To understand the development of *B*. *cinerea* on grape leaves from plants showing different resistance levels, we analyzed the two grape genotypes, ’Zi Qiu’ (HR, *V. davidii*) and ‘Riesling’ (HS, *V. vinifera*), using scanning electron microscopy (SEM). At 4 hpi, their phenotypes were approximately the same (Figure 3A,J). At 24 hpi, spore germination was clearly delayed, and fungal growth was mostly blocked on ’Zi Qiu’ leaves (Figure 3A–I). The infection rate on ‘Riesling’ leaves was more significant and destructive than that of ’Zi Qiu’ (Figure 3J–R), which *B. cinerea* failed to infect. The presence of appressoria was first noted at 18 hpi (Figure 3D) on ’Zi Qiu’, after which time the progression of infection increased slowly until 48 hpi (Figure 3D,G). At 8 hpi, appressoria were present on ‘Riesling’ (Figure 3K), while penetrations became apparent at 12 hpi (Figure 3L). The infection rate increased at 18 hpi, where infection pegs were clearly seen (Figure 3M) and again at 24 hpi, (Figure 3N), and infection hypha and necrotic spots appeared after 36 hpi (Figure 3O). From 24 hpi, fungal growth was blocked on ’Zi Qiu’, and the infection was abolished (Figure 3E). Hollow conidia, as well as some appresoria (Figure 3F), were present at 18 hpi until 96 hpi (Figure 3H,I). In contrast, fungal germination and infection was noted at 24 hpi on susceptible ‘Riesling’ leaves, which progressively increased until 96 hpi (Figure 3R). Some hyphae were branched (Figure 3Q) with apparent lesions forming. From 48 hpi onward, the fungus spread and showed sporulation on ‘Riesling’ (Figure 3P).

### 2.3. Activity of Peroxidase and Superoxide Dismutase in the HR Genotype ’Zi Qiu’ and the HS Genotype ‘Riesling’ Infected by B. cinerea

We measured the activities of superoxide dismutase (SOD) and peroxidase (POD) in the infected and control leaves. Stress conditions disrupt ROS production leading to plant cell death but plants exhibit an array of anti-oxidant enzymes to scavenge harmful ROS and protect cells from oxidative damage [19]. SOD activities in the ’Zi Qiu’ (*V. davidii*) and ‘Riesling’ (*V. vinifera*) control samples were approximately the same, except for a slightly elevated level at 4 hpi (Figure 4A) in ‘Riesling’. The activity in the inoculated ’Zi Qiu’ was approximately twice that of the control throughout the experiment (Figure 4A). The SOD activity in ‘Riesling’ was similar to that in ‘Zi Qiu’ at 4 hpi (Figure 4A), but then increased from 4 hpi to 18 hpi, where it peaked before decreasing again until 96 hpi.

POD activities in HR ’Zi Qiu’ (*V. davidii*) and HS ‘Riesling’ (*V. vinifera*) leaves were determined to evaluate the robustness of the antioxidant system during *B*. *cinerea* infection. The control samples of both genotypes, as well as inoculated ‘Riesling’, had approximately the same POD background activities over the entire time course (Figure 4B). However, in inoculated ’Zi Qiu’ leaves, POD activity increased from 4 hpi to a maximum of 48 hpi, and then decreased until 96 hpi (Figure 4B).

### 2.4. Hydrogen Peroxide (H_2_O_2_) Accumulation in HR ’Zi Qiu’ and HS ‘Riesling’ Leaves in Response to Infection with B. cinerea

Figure 5 shows H_2_O_2_ levels in leaves of ‘Zi Qiu’ (*V. davidii*) and ‘Riesling’ (*V. vinifera*) after inoculation with *B. cinerea*. The maximum H_2_O_2_ content (Figure 5A) was observed in ‘Riesling’, while the minimum H_2_O_2_ (Figure 5A) level was observed in ‘Zi Qiu’ at various time points (0, 4, 8, 12, 18, 24, 36, 48, 72, and 96 hpi). Under stress conditions, at 0 hpi, H_2_O_2_ levels were similar in ’Zi Qiu’ and ‘Riesling’ (Figure 5A). H_2_O_2_ production increased gradually in ‘Riesling’ from 4 hpi to 96 hpi, except for slight downward trend at 18 and 36 hpi. No significant increase was observed in ’Zi Qiu’compared to ‘Riesiling’ at any time point (Figure 5A) and there was no significant increase in reponse to the control treatments in either genotype.

### 2.5. Accumulation of Superoxide Radicals (O_2_^−^) in HR ’Zi Qiu’ and HS ‘Riesling’ Leaves in Response to Infection with B. cinerea

Higher O_2_^−^ (Figure 5B) levels were observed in ‘Riesling’ (*V. vinifera*) than in ‘Zi Qiu’ (*V. davidii*) at most time points. O_2_^−^ production from 0 to 4 hpi showed opposite trends in ’Zi Qiu’ and ‘Riesling’ (Figure 5B). From 8–48 hpi the levels were stable in both genotype; however, a significant increase from 48 to 96 hpi was detected in ‘Riesling’. No significant change was observed in the controls of either genotype.

### 2.6. Jasmonic Acid Levels in Leaves of the HR Genotype ‘Zi Qiu’ and the HS Genotype ‘Riesling’ Following Inoculation with B. cinerea

JA levels were determined in both ‘Zi Qiu’ (*V. davidii*) and ‘Riesling’ (*V. davidii*) samples isolated from inoculated and control treatments at various time points. JA levels were higher in ‘Zi Qiu’ (Figure 6) than in ‘Riesling’ (Figure 6) at all time points. JA levels in the ‘Zi Qiu’ control were the same as for ‘Riesling’ inoculated from 0 to 96 hpi.

### 2.7. Pearson’s Correlation Coefficients

Table 2 shows the Pearson’s correlation coefficient values of the antioxidant enzyme activities, ROS levels and JA levels in ‘Zi Qiu ’and ‘Riesling’ leaves. Significant positive correlations for SOD and POD activities with H_2_O_2_ and O_2_^−^ levels were observed. SOD activity was significantly negatively correlated with JA levels, but positively correlated with POD activity.

## 3. Discussion

Grape genotypes vary in terms of their infection resistance, degree of fungal colonization, and disease severity to *B. cinerea* [20]. Of the 81 different *Vitis* genotypes evaluated here, two were categorized as HR, twelve as resistant, twenty-five as S, and forty-two as HS (Table 1). Resistant genotypes towards *B. cinerea* have been found in *Vitis* species for example, *V. davidii*, *V. vinifera* and in the progeny of crosses between *V. vinifera* and species like *V. labrusca* (Table 1). Numerous wild Chinese *Vitis* species show multi-fungal disease resistance [21], and they have been described as important resources for future disease resistance breeding programs [22,23].

Discrete colonization of *B*. *cinerea* on grape leaves was studied by SEM at different time points. In ‘Riesling’, the pathogen initially showed limited infection, as indicated by necrosis prior to 24 hpi, but then spread substantially, and showed signs of sporulation. Prior to 24 hpi, fungal growth in ‘Zi Qiu’ leaves was significantly delayed, as evidenced by the lower germination and infection rates. Most *B. cinerea* appressoria on the ‘Zi Qiu’ leaves did not develop into infection pegs, in contrast to those on ‘Riesling’ leaves, and consistent with previous reports [11]. It was also reported that sporulation densities on *V. davidii* var. Langao-5 and *V*. *pseudoreticulata* var. Baihe-35-1 was significantly lower than those on the HS cultivar *V*. *vinifera* cv. Pinot noir [24]. Here, we saw that at the initial stages *B*. *cinerea* colonization halted in ‘Zi Qiu’ leaves.

We investigated the possible basis of differences in growth of *B*. *cinerea* in the HR genotype ‘Zi Qiu’ compared with HS ‘Riesling’. ROS is commonly produced in response to pathogen attack [25,26] and overall, higher levels of ROS accumulated in ‘Riesling’. This is not consistent with a study suggesting that in host–pathogen interactions where the pathogen is a necrotroph, pathogen-induced cell death and ROS accumulation promote pathogen growth and disease development. Thus, ROS facilitate colonization on the leaves by the necrotrophic fungus *B*. *cinerea* [27]. In contrast, low ROS levels were observed here post-inoculation in ‘Zi Qiu’, suggesting that the anti-oxidative system maintains redox equilibrium [28,26] and protects cells from ROS damage [29].

Oxidative stress disturbs the redox equilibrium in infected tissues, thereby promoting disease development [30]. In the current study, ROS accumulation after inoculation was detected in leaves from both genotypes, but with higher levels in ‘Riesling’. We conclude that ‘Riesling’ suffered significantly from the continued presence of ROS, and that ‘Zi Qiu’ did not experience substantial oxidative stress due to a high and timely anti-oxidative capacity. H_2_O_2_ higher or lower levels increase either the susceptibility or resistance respectively to *B*. *cinerea*, while, O_2_^−^ serves as a first substrate for H_2_O_2_ formation [13,30,31]. Some reports have suggested that O_2_^−^ plays a role in supporting *B*. *cinerea* invasion [32,33]. H_2_O_2_ production is induced in plant cells, accompanied by O_2_^−^ generation, which can promote programmed cell death and disease lesion development, thereby increasing *B*. *cinerea* infection [27]. We propose that the high and low levels of ROS in ‘Riesling’ and ‘Zi Qiu’ contribute to their susceptibility and resistance to *B*. *cinerea* infection, respectively [34].

We evaluated ROS accumulation and antioxidant enzyme activities during the interactions with *B*. *cinerea* [33]. Low ROS production and a timely increase in the activity of anti-oxidative enzymes were associated with the strong pathogen resistance of ‘Pingli-5’ and the HS cultivar ‘Red Globe’, which suffers from severe infection and sustained ROS production correlated with comparatively unchanged anti-oxidative activities [11]. These results support the significance of the ROS response in a timely detection of and defense against *B*. *cinerea*. We saw that the post-inoculated ‘Riesling’ leaves showed a slight variation in POD activity with lesion development. However, they showed increased SOD activity, which corresponded well with H_2_O_2_ production and a reduction in O_2_^−^. The POD activity in ‘Zi Qiu’ increased during the experiment, and no significant change was observed in SOD activity. Low levels of ROS accumulation are necessary for the anti-oxidative system to sustain redox equilibrium [26], and we also observed that the infected ‘Riesling’ showed evidence of an insufficient anti-oxidative system, resulting in consistently elevated ROS levels. In contrast, ‘Zi Qiu’ showed relatively rapid changes in anti-oxidative capacity, especially POD activity, following inoculation, and thus likely experienced less ROS-induced stress. Given that substantial levels of ROS were induced in ‘Riesling’ but not in ‘Zi Qiu’, we propose that the coordination between ROS production and scavenging mechanisms associated with the anti-oxidative system during biotic stress [35] may be a key factor in the ability of genotype ‘Zi Qiu’ to shield itself against *B*. *cinerea*.

JA levels were measured in both the HR and HS genotypes, and higher levels found in ‘Zi Qiu’. We noted high levels of JA in the ‘Zi Qiu’ control, which were approximately equal to the JA levels seen in inoculated ‘Riesling’ (Figure 5), indicating that a continuously high presence of JA in ‘Zi Qiu’ may contribute to controlling *B. cinerea*, and possibly other pathogens. These results are consistent with another study [36], which stated that high JA levels block *B*. *cinerea* infection and strengthen grape resistance to *B*. *cinerea*. Moreover, JA is known to be a major hormone involved in plant defense responses [37] and is a crucial component in the plant defense responses against insects and microbial pathogens [38]. JA accumulation occurs relatively quickly in plant tissues and cells after exposure to fungal elicitors [39,40]. JA is involved in plant response to injury and biotic stresses, such as occurs during insect and pathogen attacks [41,7], and is associated with resistance to biotrophic and necrotrophic pathogens [16,42].

## 4. Materials and Methods

### 4.1. Plant and Fungal Materials

Plant materials were obtained from the Grape Repository (34°12’N, 108°07’E) of Northwest A & F University, Yangling, Shaanxi, China. The area is situated 520 m above sea level. Mean annual temperature and precipitation are 12.9 °C and 660 mm. Most of the precipitation occurs between July and September. All the grape genotypes were cultivated in the grape germplasm depository at the Northwest A&F University, Yangling, Shaanxi, China. Fifteen plants per genotype were used for leaf evaluation. Leaves were sampled on the same dates (±3 days) in the years 2016–2017 i.e., 23 May, (genotypes 1–10), 28 May, (genotypes 11–20), 3 June, (genotypes 21–30), 8 June, (genotypes 31–40), 12 June, (genotypes 41–50), 16 June, (genotypes 51–60), 20 June, (genotypes 61–70), 25 June, (genotypes 71–81). All 81 *Vitis* genotypes collectively belong to the following grape species: (*V. vinifera* × *V. labrusca*), (*V. vinifera*), (*V. vinifera* L × *V. amurensis*), (*V. amurensis*), and (*V. davidii*). Their taxonomic information was retrieved from [43,44], and detailed information for each of the species is available in Appendix A. The material was used to evaluate gray mold disease development in 2016 and 2017.

*B. cinerea* spores were isolated from the seedless cultivar ‘Flame’ (*V*. *vinifera*) in a greenhouse located on the North campus of the Northwest A&F University, Shaanxi, China. Spores were cultured on a potato dextrose agar medium at 22 °C. After 20 days, the conidia were removed, and 1.5 × 10^6^ spores mL^−1^ were prepared in sterilized water, since this has previously been identified as the optimum inoculum [11]. The conidial suspension was confirmed to have a conidia/spore germination percentage of 95% or more before all experiments.

### 4.2. Detached Leaf Evaluation

Leaves of the same size and age (from the shoot at nodes 3 and 4) were arbitrarily selected from the grape plants. The detached leaves were washed with distilled water. For laboratory assessment, 24 leaves from each of three replicates of each genotype were evaluated. The leaves were quickly transferred to trays with 0.8% agar and sprayed evenly with the conidial suspension. Control leaves were sprayed with distilled water. The trays were placed in an incubator with a relative humidity of 90–100%, for the first 24 h in the dark and then 12/12 h light/dark at 22 °C.

### 4.3. Disease Severity Rating

Disease severity was evaluated and rated as previously described [45,46] with slight modifications. The disease symptoms observed on the leaves were ranked from 1 to 7 (Rank 1 = 0.1–5.0%, 2 = 5.1–15.0%, 3 = 15.1–30.0%, 4 = 30.1–45.0%, 5 = 45.1–65%, 6 = 65.1–85.0% and 7 = 85.1–100.0%) on the basis of the estimated percentage of lesion over the entire leaf surface. The ranking was then converted into a severity index (SI) according to the formula:$$\mathrm{SI}=\frac{\sum \left(\mathrm{Rank}\times \mathrm{number}\;\mathrm{of}\;\mathrm{infected}\;\mathrm{leaves}\;\mathrm{in}\;\mathrm{that}\;\mathrm{rank}\right)}{\mathrm{Total}\;\mathrm{number}\;\mathrm{of}\;\mathrm{leaves}\times \mathrm{highest}\;\mathrm{rank}}\times 100$$

The resistance level was rated into four classes on the basis of the SI values. Disease resistance levels of the different genotypes were categorized as highly resistant (SI: 0–1.50), resistant (SI: 1.51–3.50), susceptible (SI: 3.51–5.50), and highly susceptible (SI: 5.51–7.0). Susceptibility data for the disease were collected in 2016 and 2017. The average SI values of the two-year data were used to evaluate the resistance level.

### 4.4. Light Microscopy

Two representative genotypes from each category were used to characterize the colonization of the grape leaves by *B*. *cinerea* using light microscopy. The following genotypes were used for each category: HR, ‘Zi Qiu’ and ‘Ju Mei Gui’; R, ‘Kang San’ and ‘Rizamat’; S, ‘Flame’ and ‘Canner’; HS, ‘Riesling’ and ‘Pinot noir’. The leaves were cut into 2–3 cm^2^ segments and fixed and decolorized in 100% ethanol and in saturated chloral hydrate. The samples were stored in 50% glycerol and stained with anilineblue solution at the time of observation with an Olympus BX-51 microscope (Olympus, Tokyo, Japan) [47].

### 4.5. SEM

The development of *B*. *cinerea* on leaves of two representative genotypes, ‘Zi Qiu’ and ‘Riesling’, was observed using SEM (JEOL FESEM S-4800 scanning electron microscope, JEOL, Tokyo, Japan). Infected leaves were cut into 1.0–1.5 cm^2^ pieces at various time points (4, 8, 12, 18, 24, 36, 48, 72, and 96) hpi and immersed in 4% glutaraldehyde. After vacuum infiltration for 30 min, the infected leaves were rinsed five times for 5, 10, 15, 20, 20 min, respectively, with 0.1 M sodium phosphate buffer (PBS) (pH 6.8). The segments were dehydrated in an ethanol gradient: 30%, 50%, and 70% for 15 min each; 80% and 90% for 20 min each; and 100% alcohols twice for 30 min. Finally, the samples were incubated in acetone for 30 min and isoamyl acetate twice for 15 and 30 min in three biological replicates. The segments were desiccated by CO_2_, coated with gold in a sputter coater, and then observed under a scanning electron microscope at 15 kV [47].

### 4.6. ROS Measurement

#### 4.6.1. H_2_O_2_ Measurement

H_2_O_2_ content in ‘Zi Qiu’ and ‘Riesling’ leaves was determined at different time points (0, 4, 8, 12, 18, 24, 36, 48, 72, and 96 hpi) as previously described [48].

#### 4.6.2. O_2_^−^ Measurement

The ‘Zi Qiu’ and ‘Riesling’ O_2_^−^ production rates were calculated at different time points (0, 4, 8, 12, 18, 24, 36, 48, 72, and 96 hpi) as previously described [49]

### 4.7. Enzyme Extraction and Activity Assays

SOD activity was measured in extracts from ‘Zi Qiu’ and ‘Riesling’ leaves at different time points (0, 4, 8, 12, 18, 24, 36, 48, 72, and 96 hpi) as previously described [28]. Similarly, POD activity was measured at different time points (0, 4, 8, 12, 18, 24, 36, 48, 72, 96 hpi) as previously described [50] using about 0.5 g leaves with three biological replicates [51].

### 4.8. JA Quantification in HR and HS Genotypes

JA levels were quantified in inoculated and control ‘Zi Qiu’ and ‘Riesling’ leaves that were collected at different time points (0, 12, 24, 48, 72, and 96 hpi) and immediately frozen in liquid nitrogen. The samples were ground in liquid nitrogen with help of pestle and mortar and then stirred in 80% methanol at 4 °C overnight. The extract was methylated as previously described [52] and JA was quantified with a competitive enzyme-linked immunosorbent assay (ELISA) assay [53].

### 4.9. Statistical Analysis

Experiments were performed using three biological replicates in a randomized design. Means and standard errors were computed from independent replicates using SPSS 13.0 (SPSS Inc., Chicago, IL, USA). Least significant difference (LSD) 0.05 was employed to compute significant differences, and correlation data of the resistance evaluation from 2016 and 2017 were analyzed. All images were processed with Adobe Photoshop (Adobe Systems Incorporated, San Jose, CA, USA). All graphs were prepared using the Origin Pro 2016 32-bit software (Shenzhen, Guangdong, China) and correlation analysis was performed using the Pearson coefficient.

## 5. Conclusions

In this study, we investigated the resistance levels of different *Vitis* genotypes to *B*. *cinerea*. Most genotypes were susceptible, but a detached leaf assay revealed high resistance in clone ‘Zi Qiu’ (*V. davidii*) of *Vitis* germplasm. The results were further investigated by comparing fungal growth, ROS responses, JA levels, and changes in the antioxidant system, between the HS *V*. *vinifera* ‘Riesling’ genotype and the HR *V. davidii* ‘Zi Qiu’ genotype after *B*. *cinerea* inoculation. We observed that low ROS production, rapid elevation in antioxidant activities and high JA levels were associated with a high level of fungal resistance in ‘Zi Qiu’. In contrast, the HS genotype ‘Riesling’ suffered from severe *B*. *cinerea* infection and sustained ROS production, together with relatively unchanged anti-oxidative activities and low JA levels. This study provides insights into *B*. *cinerea* infection of grape leaves, as well as information that may be valuable to breeders in selecting germplasm for increased disease resistance.

## Figures and Tables

**Figure 1 molecules-24-00005-f001:**
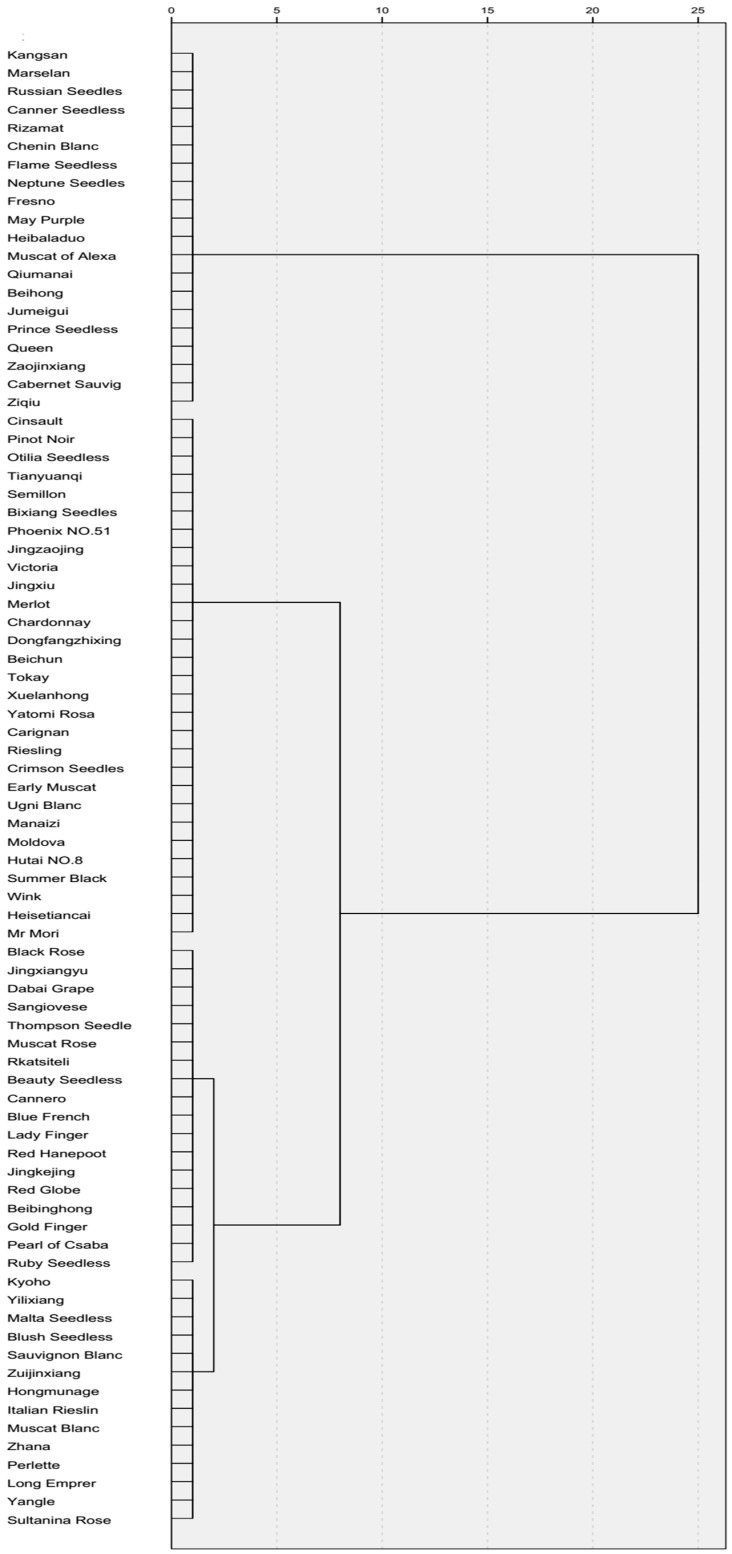
Clustering of various *Vitis* genotypes based on leaf lesion percentage, 96 h post inoculation with *Botrytis cinerea.*

**Figure 2 molecules-24-00005-f002:**
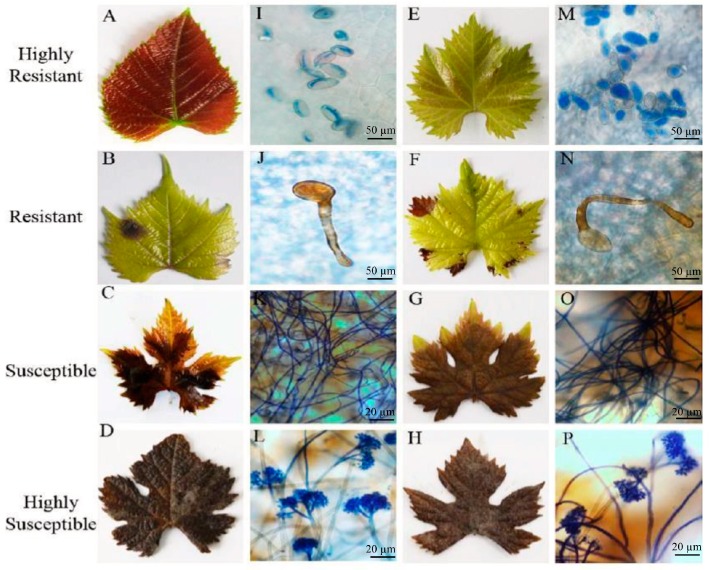
Macroscopic (**A**–**D** and **E**–**H**) and microscopic (**I**–**L** and **M**–**P**) evaluation of two representative *Vitis* genotypes, collectively representing the different *Botrytis cinerea* resistance levels. Highly resistant genotypes ’Zi Qiu’ and ‘Ju Mei Gui’ are shown in (**A**,**I**) and (**E**,**M**), respectively. Resistant genotypes ‘Kang San’ and ‘Rizamat’ are shown in (**B**,**J**) and (**F**,**N**), respectively. ‘Flame’ and ‘Canner’ represent susceptible genotypes and are shown in (**C**,**K**) and (**G**,**O**), respectively. ‘Riesling’ and ‘Pinot noir’ are highly susceptible and are shown in (**D**,**L**) and (**H**,**P**), respectively. Scale bars (**I**,**J**,**M**,**N**): 50 µm; (**K**,**L**,**O**,**P**): 20 µm. Samples were collected 96 h after inoculation and then one representative leaf of three biological replicates were analyzed.

**Figure 3 molecules-24-00005-f003:**
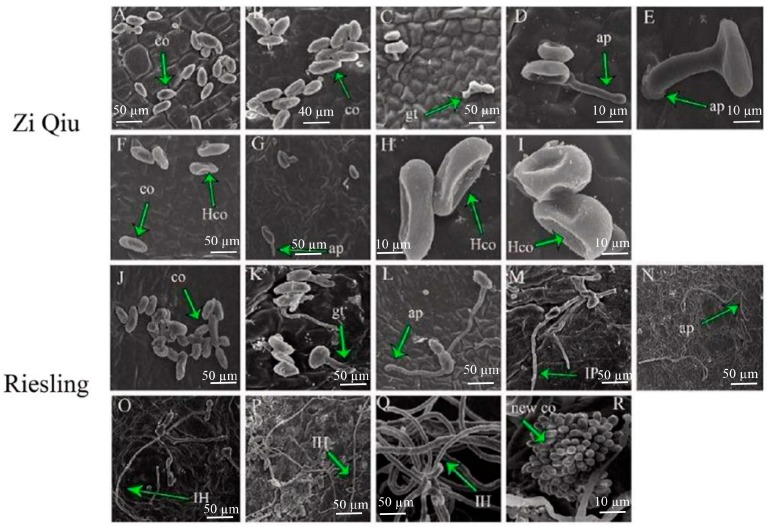
Comparison of *Botrytis cinerea* conidia development on ’Zi Qiu’ and ‘Riesling’ leaves using scanning electron microscopy. Progression of *B*. *cinerea* colonization on ’Zi Qiu’ (**A**–**I**) and ‘Riesling’ (**J**–**R**). Leaves were harvested at 4, 8, 12, 18, 24, 36, 48, 72, and 96 hpi, and the experiments were repeated three times. Arrows indicate co, conidia; gt, germ tube; ap, appressoria; IP, infection peg; IH, infection hypha; new co, new conidia; and Hco, hollow conidia.

**Figure 4 molecules-24-00005-f004:**
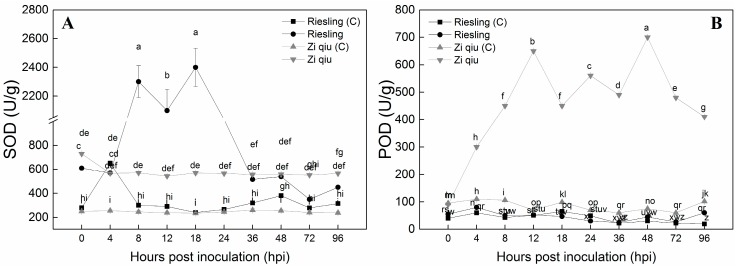
Superoxide dismutase (SOD) (**A**) and peroxidase (POD) (**B**) activities of protein extracts from ’Zi Qiu’ and ‘Riesling’ leaves at 0, 4, 8, 12, 18, 24, 36, 48, 72, and 96 h post-inoculation (hpi) with *Botrytis cinerea*, using sterile water as the control. Three independent experiments were used for the means and standard errors. Small alphabetes indicate significant differences according to LSD test (*p* < 0.05) between “Zi qiu” and “Riesling”.

**Figure 5 molecules-24-00005-f005:**
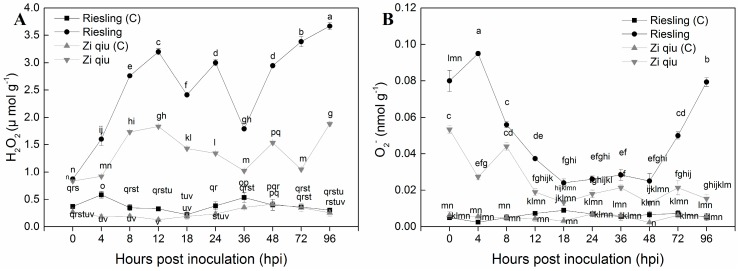
Levels of H_2_O_2_ (**A**) and O_2_^−^ (**B**) in leaves of highly resistant ’Zi Qiu’ and highly susceptible ‘Riesling’ at 0, 4, 8, 12, 18, 24, 36, 48, 72, and 96 hpi with *Botrytis cinerea* and using sterile water as the control. Three independent experiments were used for the means and standard errors. Small alphabetes indicate significant differences according to LSD test (*p* < 0.05) between “Zi qiu” and “Riesling”.

**Figure 6 molecules-24-00005-f006:**
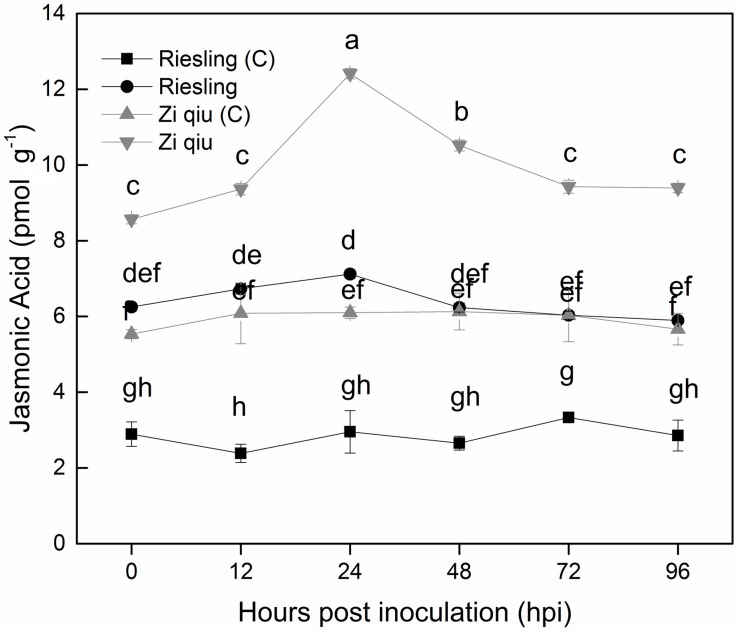
Jasmonic acid (JA) levels in highly resistant ‘Zi Qiu’ and highly susceptible ‘Riesling’ leaves at 0, 12, 24, 48, 72, and 96 hpi with *Botrytis cinerea* and using sterile water as the control. Three independent experiments were used for the means and standard errors. Small alphabetes indicate significant differences according to LSD test (*p* < 0.05) between “Zi qiu” and “Riesling”.

**Table 1 molecules-24-00005-t001:** Laboratory evaluation of disease severity in 81 *Vitis* genotypes, following inoculation with *Botrytis cinerea* in 2016 and 2017.

Species	Name of Genotype	Disease Severity ^a^ (%)	Resistance Level ^b^	Geographical Origin
*V. vinifera* L.	Beauty Seedless	83.6	HS	University of California, Davis, CA, USA
*V. vinifera* L. × *V. amurensis* Rupr	Beibinghong	84.6	HS	1995, Institute of Special Animal and Plant Science of CAAS, Beijing, China
*V. vinifera* L. × *V. amurensis* Rupr	Beichun	93.6	HS	1954, Beijing Botanical Garden Institute of Botany, Chinese Academy of Science, Beijing, China
*V. vinifera* L. × *V. amurensis* Rupr	Beihong	42.4	S	1954, Beijing Botanical Garden Institute of Botany, Chinese Academy of Science, China
*V. vinifera* L.	Bixiang Seedless	92.2	HS	1994, Jilin Academy of Agricultural Science, Changchun, China
*V. vinifera* L.	Black Rose	76.5	S	The Division of Horticultural Research of Commonwealth Scientific and Industrial Research Organization (CSIRO), Melbourne, Australia
*V. vinifera* L.	Blue French	76.5	S	Austria
*V. vinifera* L.	Blush Seedless	71.8	S	Professor Olmo, Davis Station, CA, USA.
*V. vinifera* L.	Cabernet Sauvignon	22.7	R	Aquitaine, Bordeaux, France
*V. vinifera* L.	Canner Seedless	40.5	S	University of California, Davis, CA, USA.
*V. vinifera* L.	Cannero	80.5	HS	Cannero Riviera, Italy
*V. vinifera* L.	Carignan	93.7	HS	Carinena, Aragon, Spain
*V. vinifera* L.	Chardonnay	96.1	HS	Burgundy, France
*V. vinifera* L.	Chenin Blanc	27.4	R	Anjou, Loire Valley, France
*V. vinifera* L.	Cinsault	91.2	HS	Languedoc, France
*V. vinifera* L.	Crimson Seedless	94.2	HS	Institute of Fruit Tree Research of Changli, Hebei Academy of Agriculture and Forestry Science, Shijiazhuang, China
*V. vinifera* L.	Dabai Grape	73.2	S	Unknown
*V. vinifera* L. × *V. labrusca* L.	Dong fang zhixing	94.3	HS	2007, Hiroshima, Japan
*V. vinifera* L.	Early Muscat	98.6	HS	1997, Shandong Province, Jinan, China
*V. vinifera* L.	Flame Seedless	45.8	S	1973, Freson, CA, USA
*V. vinifera* L.	Fresno Seedless	51.7	S	Fresno, CA, USA
*V. vinifera* L. × *V. labrusca* L.	Gold Finger	83.6	HS	1982, Japan
*V. vinifera* L. × *V. labrusca* L.	Hei bala duo	39.2	R	2004, Kumamoto Prefecture, Japan
*V. vinifera* L.	Heise tiancai	93.0	HS	2009, Kofu, Japan
*V. vinifera* L.	Hongmu nage	58.7	S	Atux, Xinjiang, China
*V. vinifera* L. × *V. labrusca* L.	Hutai NO.8	93.6	HS	Grape Institute, Xian, China
*V. vinifera* L.	Italian Riesling	59.9	S	Italy
*V. vinifera* L.	Jing xiu	92.0	HS	1994, Chinese Academy of Science, Beijing, China
*V. vinifera* L.	Jing ke jing	81.2	HS	1984, Chinese Academy of Science, Beijing, China
*V. vinifera* L.	Jingzaojing	91.8	HS	1984, Chinese Academy of Science, Beijing, China
*V. vinifera* L.	Jinxiangyu	78.5	HS	1997, Chinese Academy of Science, Beijing, China
*V. vinifera* L.	Ju mei gui	4.6	HR	Dalian Academy of Agriculture Science, Dalian, China
V. riparia L. × *V. labrusca* L.	Kang san	18.0	R	Unknown
*V. vinifera* L.× *V. labrusca* L.	Kyoho	25.7	R	1937, Japan
*V. vinifera* L.	Lady Finger	87.6	HS	1984, Japan
*V. vinifera* L.	Long Emprer	48.8	S	Unknown
*V. vinifera* L.	Malta Seedless	73.6	S	Malta
*V. vinifera* L.	Manaizi	95.2	HS	Tulufan, Xinjiang, China
*V. vinifera* L.	Marselan	24.6	R	France
*V. vinifera* L.	May Purple	38.0	R	Unknown
*V. vinifera* L.	Merlot	94.5	HS	Bordeaux, France
*V. vinifera* L.	Moldova	95.3	HS	Moldova
*V. vinifera* L.	Mr Mori	94.5	HS	1985, Japan
*V. vinifera* L.	Muscat Blanc	63.4	S	Eastern Mediterranean
V. vinifera L	Muscat of Alexandia	42.0	S	Egypt
*V. vinifera* L.	Muscat Rose	78.1	HS	Greece
*V. vinifera* L.× *V. labrusca* L.	Neptune Seedless	47.2	S	1998, University of Arkansas, AR, USA.
*V. vinifera* L.	Otilia Seedless	93.2	HS	Romania
*V. vinifera* L.	Pearl of Csaba	85.3	HS	1904, Hungary.
*V. vinifera* L.	Perlette	59.5	S	Fresno, CA, USA.
*V. vinifera* L.	Phoenix NO.51	88.5	HS	Germany
*V. vinifera* L.	Pinot Noir	94.0	HS	Burgundy, France
*V. vinifera* L.	Prince Seedless	23.4	R	Hebei academy of agriculture and forestry science, Changli, China
*V. vinifera* L.	Qiu manai	41.6	S	Atushi, Xinjiang, China
*V. vinifera* L.	Queen	24.6	R	Unknown
*V. vinifera* L.	Red Globe	85.0	HS	University of California, CA, USA
*V. vinifera* L.	Red Hanepoot	82.1	HS	Unknown
*V. vinifera* L.	Riesling	98.0	HS	Germany
*V. vinifera* L.	Rizamat	27.3	R	The Soviet Union
*V. vinifera* L.	Rkatsiteli	75.0	S	Georgia
*V. vinifera* L.	Ruby Seedless	86.3	HS	University of California, CA, USA.
*V. vinifera* L.	Russian Seedless	27.2	R	Australia
*V. vinifera* L.	Sangiovese	78.1	HS	Italy
*V. vinifera* L.	Sauvignon Blanc	71.4	S	France
*V. vinifera* L.	Semillon	93.2	HS	France
V. vinifera L	Sultanina Rose	54.6	S	Unknown
*V. vinifera* L. × *V. labrusca* L.	Summer Black	97.6	HS	1968, Yamanashi Prefecture, Japan
*V. vinifera* L.	Thompson Seedless	76.2	S	Tulufan, Xinjiang, China
*V. vinifera* L. × *V. labrusca* L.	Tian yuan qi	90.0	HS	1994, Fruit Insitute, Liaoning, China
*V. vinifera* L.	Tokay	93.9	HS	Hungary
*V. vinifera* L.	Ugni Blanc	97.6	HS	Italy
*V. vinifera* L.	Victoria	92.0	HS	Romania
*V. vinifera* L.	Wink	93.4	HS	1998, Japan
*V. amurensis* Rupr	Xue lanhong	94.4	HS	Institute of Special Animal and Plant Science of CAAS, Jilin, China
*V. vinifera* L.	Yangle	55.8	S	Russia
*V. vinifera* L.	Yatomi Rosa	95.2	HS	1990, Japan
*V. vinifera* L.	Yili xiang	69.8	S	Yili, Xinjiang, China
*V. vinifera* L. × *V. labrusca* L.	Zao jin xiang	28.2	R	1963, Grape institute, Liaoning, China
*V. vinifera* L.	Zhana	63.8	S	Albania
*V. davidii* Foex	Zi qiu	3.3	HR	2004, Hunan Agricultural University, Hunan, China
*V. vinifera* L. × *V. labrusca* L.	Zui jin xiang	67.8	S	1997, Liaoning Academy of Agriculture Science, China

^a^ Disease severity: the average percentage of spreading lesions determined by observation of at least 24 leaves in each 2016 and 2017 experiment. ^b^ Resistance level: Highly Resistant (HR: rank of 0–1.50); Resistant (R: rank of 1.51–3.50); Susceptible (S: rank of 3.51–5.50); Highly Susceptible (HS: rank of 5.51–7.0).

**Table 2 molecules-24-00005-t002:** Pearson’s correlation coefficients of antioxidant enzyme activities, with reactive oxygen species (ROS) and jasmonic acid (JA) contents in ‘Zi Qiu’ and ‘Riesling’ leaves.

	JA	O_2_^−^	H_2_O_2_
SOD	−0.9959 *	0.8182 *	0.8974 *
POD	0.0831	0.5278 *	0.3975 *

SOD, superoxide dismutase; POD, peroxidase; H_2_O_2_, hydrogen peroxide; O_2_^−^, superoxide radicals; JA, jasmonic acid. * Significant at the 0.01 probability level.

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
