# Peer review of "Screening Vitis Genotypes for Responses to Botrytis cinerea and Evaluation of Antioxidant Enzymes, Reactive Oxygen Species and Jasmonic Acid in Resistant and Susceptible Hosts"

_molecules, 2018, doi:10.3390/molecules24010005_

Reviewer 1 Report

The paper has improved since last submission.

Author Response

Thank you very much for your kind helps. We have revised our manuscript according to editor and reviewer's suggestions.

Reviewer 2 Report

English revision is not satisfactory. Also, lots of typing errors must be corrected. Conversely to what stated in the rebuttal letter, some comments of the previous review round have not been addressed, and no reasons have been reported for that. In the figures, the caption should report information on the statistics; also, several points lack statistics letters. It is very difficult for me to track several other corrections, because line numbering in the rebuttal letter is incorrect.

Author Response

Reviewer # 2

Comments and Suggestions for Authors

(1) English revision is not satisfactory

Response: Thank you very much for your valuable suggestions. We have invited Prof. and Dr.

Jocelyn Rose (Plant Biology Section, School of Integrative Plant Science Director, Cornell

University) to help, edit and improve our revised manuscript.

(2) Also, lots of typing errors must be corrected.

Response: We apologize for the typos mistakes. We revised the manuscript and necessary corrections were incorporated.

(3) Conversely to what stated in the rebuttal letter, some comments of the previous review round have not been addressed, and no reasons have been reported for that.

Response: Thank you for pointing out some issues. We tried to address all the comments but if some comments were not answered unintentionally then we apologize for that. According to your suggestions we carefully revised previous comments again and edited our manuscript. If some comments of the previous review round have not been addressed, we must redo it.

(4) In the figures, the caption should report information on the statistics; also, several points lack statistics letters.

Response: Thank you very much for valuable suggestions. We have added statistical information in captions (Line number: 156, 184 and 194) and statistical letters were added to all points (Figure 4, Figure 5 and Figure 6).

(5) It is very difficult for me to track several other corrections, because line numbering in the rebuttal letter is incorrect.

Response: Thank you for your suggestion. We apologize for the unintentional mix up of line numbering in manuscript and rebuttal letter of previous review. We also have carefully revised previous comments and edited our manuscript.

Reviewer 3 Report

1. There are also many errors in the paper, most are typos (e.g. no spaces between several words); format of tables still not corrected

2. The use of the terms (resistance and susceptibility) is not consistently; sometimes the author use it for the plant (as in title) other times for the fungus as in the abstract. Check across the manuscript

3. It is better to use the (gentype name) with the scientific name of species in the resuults section, because there are different Vitis species. e.g. the use of (Riesling) in the section of (antioxidant aspect).

4. The main point in my previous review was about (Similar work to this manuscript have been published in other reference [4]. There are no big differences with the new manuscript vision.

a. Both works reported: [wild Chinese grape (Vitis species) ] showed high resistance HR against Botrytis cinerea]. However, in the reference work [4], the wild species was a non-identified species (Chinese wild Vitis sp.); while in this manuscript is (Chinese wild Vitis davidii). 

b. Both works reported: genotype (Vitis davidii) show high sensible HS to the fungus.

5. Conclusion section lacks. 

5.       Therefore, I found that the subject data would still be insufficient to publish.

Ref [4] 

    Wan, R.; Hou, X.; Wang, X.; Qu, J.; Singer, S.D.; Wang, Y.; Wang, X. Resistance evaluation of Chinese wild                 Vitis genotypes against Botrytis cinerea and different responses of resistant and susceptible hosts to the infection.         Front. Plant Sci. 2015, 6, 1-17

Author Response

Reviewer # 3

Comments and Suggestions for Authors

1. There are also many errors in the paper most are typos (e.g. no spaces between several words); format of tables still not corrected

Response: We apologize for the typos mistakes. We revised the manuscript and necessary corrections were incorporated according to your suggestions. The table format was changed according to the comments and suggestions of other reviewers in the previous round of review.

2. The use of the terms (resistance and susceptibility) is not consistently; sometimes the authors use it for the plant (as in title) other times for the fungus as in the abstract. Check across the manuscript

Response: Thank you very much for your valuable suggestions. We revised this in our revised

Manuscript (Line: 20).

3. It is better to use the (genotype name) with the scientific name of species in the results section, because there are different Vitis species. e.g. the use of (Riesling) in the section of (antioxidant aspect).

Response: we agree to your comment and added species name following the scientific names in the result section (Line: 120, 121, 146, 159, 166, 176 and 188).

4. The main point in my previous review was about (Similar work to this manuscript have been published in other reference [4]. There are no big differences with the new manuscript vision.

a. Both works reported: [wild Chinese grape (Vitis species)] showed high resistance HR against Botrytis cinerea]. However, in the reference work [4], the wild species was a non-identified species (Chinese wild Vitis sp.); while in this manuscript is (Chinese wild Vitis davidii). 

b. Both works reported: genotype (Vitis davidii) show high sensible HS to the fungus.

Response: Thank you very much for valuable suggestions. We make it clear in the revised version of our revised manuscript at Line # 55-60 about the novelty of work. In the reference work [4], the authors have worked on Chinese wild Vitis species while in our study we included various genotypes i.e., eleven (V. vinifera × V. labrusca), sixtyfive (V. vinifera), three (V. vinifera L × V. amurensis genotypes), one (V. amurensis) and one clone of (V. davidii) (Total 81 materials) which were not included in the previous study. As it was the part of the same project of our Laboratory, thus we included more Vitis genotypes present in the same repository for evaluation.

5. Conclusion section lacks. 

Response: According to Molecules format, the conclusion section is optional so that’s why we didn’t keep it separate, but now we followed your suggestion and split the conclusion section separately in our revised manuscript (Line: 358-369).

 Round  2

Reviewer 2 Report

It is frustrating to see again that language and typing errors still persist. Typing errors still persist. Some errors are reported below, but they are just some examples. If a colleague revised the manuscript, he should be cited in the acknowledgments.

L17: screening

L23: inoculation

L68-70: revise the sentence

L156: Means and standard errors of three replicates are reported. Means with different letters are significantly different according to LSD test (P<0.05). Do the same corrections for the other figure captions.

L356: and correlation analysis was performed using the Pearson coefficient.

L209-210: infection resistance????? To what?????? Re-write.

L211-212: Resistant genotypes to B. cinerea were found in ...

L214: delete 'an'

L218: and showed

L224: cinerea

Author Response

Reviewer # 2

Comments and Suggestions for Authors

It is frustrating to see again that language and typing errors still persist. Typing errors still persis.Some errors are reported below, but they are just some examples. If a colleague revised the manuscript, he should be cited in the acknowledgments.

Response: Thank you very much for your valuable suggestions. We apologize for the typos mistakes. We revised the manuscript and necessary corrections were incorporated. We revised our manuscript by Prof. Jocelyn Rose for English editing and thanked Plant Scribe in the acknowledgement section on his recommendation (http://www.plantscribe.com/). But according to your suggestions we replaced the “Plant Scribe” with the name of “Prof. Jocelyn Rose” Line No. 380.

Corrections have been done in the following lines:

L17: screening

Response: We replaced the word “screen” with “screening” (Line number: 17).

L23: inoculation

Response: We replaced the word “infection” with “inoculation” (Line number: 23).

L68-70: revise the sentence

Response: We revised the sentence according to your suggestion (Line number: 68-69).

L156: Means and standard errors of three replicates are reported. Means with different letters are significantly different according to LSD test (P<0.05). Do the same corrections for the other figure captions.

Response: We revised the sentences according to your suggestion (Line number: 156, 184 and 196).

L356: and correlation analysis was performed using the Pearson coefficient.

Response: We revised the sentence and added the suggested line (Line number: 356).

L209-210: infection resistance????? To what?????? Re-write.

Response: We rewrote the sentence according to your suggestion (Line number: 209).

L211-212: Resistant genotypes to B. cinerea were found in ...

Response: We modified the sentence according to your suggestion (Line number: 211).

L214: delete 'an'

Response: We deleted “an” in the sentence according to your suggestion (Line number: 213).

L218: and showed

Response: We added “and” in the sentence according to your suggestion (Line number: 217).

L224: cinerea

Response: We replaced “C” with “c” in the sentence according to your suggestion (Line number: 224).

Moreover, in the light of your valuable suggestions we also made corrections in the text (track changes) regarding space or typos mistakes in the following lines: (Line number: 117, 124, 135, 140, 147, 149, 153, 154, 163, 173, 181, 182, 193, 223, 280, 281, 282, 324, 337 and 349).

Reviewer 3 Report

I find that there is no different between the results of this work and the previous work [4]. Same Vitis species showed the same results even the many Vitis species were tested in this present work.  The answer about this point is not enough, there are only more Vitis species and genotypes that used but the results are same. No discuss about this in discussion section. Therefore, the novel and new results in this work lack.

Author Response

Reviewer # 3 Comments and Suggestions for Authors

I find that there is no different between the results of this work and the previous work [4]. Same Vitis species showed the same results even the many Vitis species were tested in this present work.  The answer about this point is not enough, there are only more Vitis species and genotypes that used but the results are same. No discuss about this in discussion section. Therefore, the novel and new results in this work lack.

Response: Thank you very much for valuable suggestions. We make it clear in the revised version of our article at Line # 55-60 about the novelty of work. In the reference work [4], the authors have worked on Chinese wild Vitis species while in our study we also included various genotypes i.e., eleven (V. vinifera × V. labrusca), sixty-five (V. vinifera), three (V. vinifera L × V. amurensis genotypes), one (V. amurensis) and one clone of (V. davidii) (Total 81 materials) which were not included in the previous study. As it was the part of the same project of our Laboratory, thus we included more vitis genotypes present in the same repository for evaluation.